Unified extractive-abstractive summarization: a hybrid approach utilizing BERT and transformer models for enhanced document summarization

S. Divya 1 divya_stephenson@yahoo.com
N. Sripriya 1
http://orcid.org/0000-0003-3592-6543 Andrew J. 2 andrew.j@manipal.edu
http://orcid.org/0000-0002-3860-4948 Mazzara Manuel 3
1 Department of Information Technology, SSN College of Engineering , Kalavakkam, Tamil Nadu , India
2 Department of Computer Science and Engineering, Manipal Institute of Technology, Manipal Academy of Higher Education , Manipal, Karnataka , India
3 Institute of Software Development and Engineering, Innopolis University , Innopolis , Russia
Lara Juan
Electronic publication date: 2024 Nov 18
Publication date: 2024
Volume: 10
Electronic Location ID: e2424
Received 2024 Jun 14; Accepted 2024 Sep 26
Copyright: © 2024 S et al.
Copyright year: 2024
Copyright holder: S et al.
License: This is an open access article distributed under the terms of the Creative Commons Attribution License, which permits unrestricted use, distribution, reproduction and adaptation in any medium and for any purpose provided that it is properly attributed. For attribution, the original author(s), title, publication source (PeerJ Computer Science) and either DOI or URL of the article must be cited.
License URL: https://creativecommons.org/licenses/by/4.0/

Keywords: Document summarization, BERT, CNN, Transformer models, Abstractive summarization

Funding: The authors received no funding for this work.

==============================
With the exponential proliferation of digital documents, there arises a pressing need for automated document summarization (ADS). Summarization, a compression technique, condenses a source document into concise sentences that encapsulate its salient information for summary generation. A primary challenge lies in crafting a dependable summary, contingent upon both extracted features and human-established parameters. This article introduces an intelligent methodology that seamlessly integrates extractive and abstractive techniques to ensure heightened relevance between the input document and its summary. Initially, input sentences undergo transformation into representations utilizing BERT, subsequently transposed into a symmetric matrix based on their similarity. Semantically congruent sentences are then extracted from this matrix to construct an extractive summary. The transformer model integrates an objective function highly symmetric and invariant under unitary transformation for language generation. This model refines the extracted informative sentences and generates an abstractive summary akin to manually crafted summaries. Employing this hybrid summarization technique on the CNN/DailyMail dataset and DUC2004, we evaluate its efficacy using ROUGE metrics. Results demonstrate the superiority of our proposed technique over conventional summarization methods.

Introduction

The rapid expansion of web-based documents underscores the importance of extracting subject-based context from extensive texts, constituting a crucial research domain within natural language processing (NLP). This endeavor, commonly referred to as text summarization (Reategui, Klemann & Finco, 2012), entails the generation of summaries varying in size depending on both the content of the source document and user preferences. Traditional classifications of text summarization (Rahimi, Mozhdehi & Abdolahi, 2017; Rush, Chopra & Weston, 2015) delineate between extraction and abstraction techniques. Extractive summarization involves identifying and selecting informative sentences for summary creation, while abstractive summarization entails synthesizing information from the source document, rephrasing it, and generating a summary akin to those crafted manually.

Over recent decades, text summarization has encountered numerous challenges, prompting the development of solutions dating back to the 1950s. The initial approach involved extracting features from text documents, a pivotal step in generating standard summaries. Initially, emphasis was placed on factors such as word and phrase frequency to discern important sentences. Subsequent advancements expanded the array of features considered, encompassing sentence position, part of speech, word frequency, and sentence length (Nallapati et al., 2016), facilitating the identification of informative sentences. Furthermore, novel methodologies emerged, including the application of vector space representation to map text onto a vector space. Techniques such as matrix decomposition, dimensionality reduction, and deriving sentence similarity within the document facilitated the extraction of informative content (Conroy et al., 2013; He et al., 2018). Graph-based summarization techniques adopted a framework wherein each sentence’s elements represented nodes, and the relationships among these nodes contributed to summary generation (Mihalcea & Tarau, 2004). Moreover, the combinational optimization approach sought to achieve optimal solutions by employing techniques such as integer linear programming, submodular functions, and combinatorial optimization to ensure coverage and diversity in summarization (Popescu, Grama & Rusu, 2021).

The process of abstractive summarization involves the potential inclusion of terms and phrases not explicitly present in the original document. This method relies on encoder-decoder or sequence-to-sequence models for its generation. Recently, novel abstractive summarization techniques have emerged as solutions to address the Out Of Vocabulary (OOV) problem. Notable among these are Copy-Net’s network structure (Gu et al., 2016), which employs end-to-end training, and Ptr-Net (Vinyals, Fortunato & Jaitly, 2015), a framework incorporating encoder-decoder phases within a recurrent neural network (RNN). These models tackle the OOV challenge through a two-phase approach: the generation phase and the copy phase. This dual mechanism enables the production of diverse words by implementing a state update mechanism.

However, abstractive content generation often encounters challenges such as poor legibility, data duplication, and significant deviation from the source content (Cao et al., 2018; Yang et al., 2019). Consequently, the resulting summary may not faithfully represent the exact content of the source document. Conversely, while extractive summarization methods may suffer from bias and limited coverage (Vaswani et al., 2017), they still struggle to capture the overall essence of the source material. The proposed hybrid summarization model aims to simultaneously achieve three objectives: coverage, relevance, and diversity. To evaluate its performance, this hybrid technique is applied to the CNN/DailyMail and DUC2004 dataset and assessed using ROUGE metrics. The results demonstrate the superiority of the proposed approach over standard summarization techniques.

Key contributions

The key contributions are as follows: An efficient intermediate representation for each sentence in the document is generated by extracting semantical features using an embedding technique.

A hybrid approach using both extractive and abstractive summarization techniques for generating concise and informative summary is proposed.

For extractive summarization, a contextually firm system is developed that learns the context of the document end-to-end to extract the informative sentences eliminating the redundancy.

An abstractive summarizer is developed that takes the informative sentences obtained from the extractive summarizer to generate coherent summary that are similar to the human generated summary. The rest of this article is organized as follows: “Related Works” provides a brief overview of existing extractive and abstractive summarization techniques. “Methodology” details the methodology of the proposed hybrid summarization technique. “Experiments and Results” outlines the experimental setup and presents the results. Finally, “Conclusion and Future Works” concludes the article and discusses avenues for future research.

Related works

The objective of the automatic document summarization task entails the relevant information to generate a summarized document. Several summary generation methods have evolved and have rendered support for an effective processing of huge documents. Based on the linguistic level, summarization is categorized as either abstractive or extractive (Gambhir & Gupta, 2017). Earlier techniques are mostly centered on extractive summarization. Generally, a primary sentence that represents the content available in the document is examined based on its position in the input source (Afsharizadeh, Ebrahimpour-Komleh & Bagheri, 2018), frequent occurrence of the term (Sakhadeo & Srivastava, 2018), and sentence hierarchy based on the amount of information available (Narayan, Cohen & Lapata, 2018a). At the lexical level, text modelling is performed using n-grams. For example, in Ledeneva, García-Hernández & Gelbukh (2014) model, the maximal frequent sequences method is applied to retrieve n-gram sequences from the text input. In contrast, recognition of n-grams is performed to detect the primary terms for constructing a paragraph in the document (Bando et al., 2007). Extracted features are handled by supervised and unsupervised techniques to build models that facilitate the identification of primary components of fundamental concepts.

Generally, summarization techniques are categorized into two approaches namely, supervised or unsupervised. With respect to supervised approaches, many classification tasks include input structure either labelled as “1” or “0”. Those sentences present in the summary are “1” and remaining labelled as “0”. Diverse features are fetched from each sentence by implementing statistical and linguistic-focused methods. The sentences are categorized based on features using a standard classification algorithm (Bando et al., 2007). Another kind of supervised method is developed in which the model is trained by several extracted features. The connection among features is examined to assign weights to each sentence. This weight allocation is obtained by a genetic algorithm (GA) (Rojas Simón, Ledeneva & García-Hernández, 2018) and a regression model (Vázquez, Ledeneva & García-Hernández, 2019). The major challenge with supervised approach is that it necessitates a set of labelled data. Additionally, the processing of training samples is generally not common for multiple domains.

Recently, summarization techniques are classified into three categories namely, graph-based (Cengiz et al., 2018), neural-network-based and clustering-based (Tan, Wan & Xiao, 2017; Yang, Li & Zhao, 2019). In a graph-based model (Cengiz et al., 2018), each sentence or term is handled based on a graph data structure (Rao et al., 2020). This method is centered on detecting significant sentences from a document. Fundamentally, the prominence of a vertex within a graph is assessed. Generally, in text-based ranking, a uni-directed and weighted graph is created. Documents or sentences are comprised of nodes. Nodes that hold relevant information are connected by edges. Based on the level of semantic relevancy between nodes, weights are allocated to each edge and thus facilitate sentence scoring. The nodes that are connected with edges having greater weights are ranked high and are considered as summary sentences.

In clustering-based model (Tan, Wan & Xiao, 2017), a group of sentences holding semantically similar content are identified. Several features are extracted from each input sentence. Sentences having similar features are grouped to form a cluster. Major limitation in cluster generation is that the content with the same clusters has to be tightly packed whereas the content within different clusters is loosely packed. To ensure a quality summary, an evaluation of these groups has to be performed.

In a neural-network based model, due to an extensive development of deep learning and availability of data, wide research has been carried out on document-level summarization. Deep learning-based summarization techniques use sequence models such as RNN, LSTM, BERT, etc., for processing. Later, attention mechanisms (Gehrmann, Deng & Rush, 2018), fine-grained (Li et al., 2020), and SummaRuNNer (Liu, 2019) are used for processing the input document to generate a summary.

Pre-trained language models

Recent advancements in natural language processing (NLP) have seen the development of several language models (Nallapati, Zhai & Zhou, 2017; Peters et al., 2018; Radford et al., 2018; Devlin et al., 2018; Dong et al., 2019), resulting in significant progress across various NLP tasks. These models have expanded the scope of embeddings by extracting semantic representations from extensive datasets using language modelling objectives. One prominent model in this domain is Bidirectional Encoder Representations from Transformers (BERT) (Radford et al., 2018), designed for masked language modelling (MLM) and next sentence prediction (NSP) tasks, trained on a dataset comprising 3,300 million words. BERT is sensitive to word order and incorporates symmetric regularization into its learning process.

The preprocessing of input text involves the insertion of specific tokens at the beginning and end of each sentence. The comprehensive information from the entire input sequence is captured through output representations, with the token at the beginning and end signifying the boundaries of a sentence. This processed text is represented as a sequence of tokens, with each token characterized by token embedding, segmentation embedding, and position embeddings. Token embedding captures the meaning of each token, segmentation embeddings distinguish between two sentences, and position embeddings denote the position of each token in the input sequence. These embeddings are aggregated into a single vector and passed through multiple layers in the bidirectional transformer.

The bidirectional transformer model generates contextual representations, utilizing multiple attention heads for each input sequence. This representation is expressed as:

(1) e~l=LN(e+MAtt(el−1))

(2) el=LN(e~l+FFN(e~l))

where, e0=x are the input vectors, LN is the normalization function for each layer, Multiple head attention operation is representaed as MATT, a double layer feed forward neural network operation is given as FFN. l represents the depth of the stacked layer.

Output vector obtained from BERT comprises rich context-sensitive information. Such a pre-trained model serves to upgrade the effectiveness of language understanding tasks. Currently, pre-trained models are imposed in several generation challenges (Rothe, Narayan & Severyn, 2020; Zhang et al., 2018). While fine-tuning for a definite task, unlike other traditional models like ELMo (has fixed parameters), parameters are collectively fine-tuned with supplemental task-oriented parameters.

Extractive summarization

Summary generation is done by identifying the most significant sentences in the document using the extractive summarization technique. Neural network-based summarization model handles the extractive summarization task as a sentence classification problem. Neural encoder phase produces representations for each sentence and classifier identifies the sentences that have to be included in the summary. The earliest neural network approach for extractive summarization is SummaRuNNer (See, Liu & Manning, 2017) which enacts an RNN-based encoder. A reinforcement learning-based summarization model is Refresh (Narayan, Cohen & Lapata, 2018a) in which training is performed by global optimization of the ROUGE evaluation metric. Latent model (Zhang et al., 2018) projects extractive summarization as an inference problem with latent variables that directly optimizes the susceptibility of human summaries given selective summary sentences rather than just optimizing the susceptibility of assigned labels. In a model termed Sumo (Liu, Titov & Lapata, 2019) multi-root dependency tree representation of the source document is provoked with an idea of structured attention during the process of identifying the output summary. A state-of-the-art model in extractive summarization termed NeuSum (Zhou et al., 2018) concurrently scores and picks sentences for generating a summary.

Abstractive summarization

Neural network-based approaches address the abstractive summarization task as a problem that has to be handled one sequence after the other. Encoder phase corresponds to the series of tokens in the input document t={t1,t2,…,tn}to a serious of consecutive representations z={z1,z2,…,zn}, whereas the decoder phase, produces the summary y={y1,y2,…,ym}token-by-token in an auto-regressive fashion, therefore modelling the contingent probability p(y1,y2,…,ym|t1,t2,…,tn).

Rush, Chopra & Weston (2015) and Nallapati et al. (2016) proposed a model that implemented the encoder-decoder architecture for summarization. An improvement is performed in this model, PTGEN (See, Liu & Manning, 2017) is proposed by applying a pointer generator network that admits the reproduction of words from the input text and a Cov (coverage) mechanism for tracing words towards summarization. An abstractive summarization model (See, Liu & Manning, 2017) is proposed where a hierarchical attention mechanism facilitates numerous factors to collectively represent the documents for decoding. End-to-end training is performed on their deep communication agent (DCA) using reinforcement learning. An abstractive summarization approach (Celikyilmaz et al., 2018) that incorporates Deep Reinforcement Model (DRM) utilizes intra-attention mechanism to solve the coverage problem. BOTTOMUP (Gehrmann, Deng & Rush, 2018) approach for summarization incorporates content selector that initially identifies the phrases from the source document to be included in the summary and decoder handles the preselected phrases alone with copy mechanism. An extremely abstractive summarization (single sentence summary) model (TConvS2S) (Narayan, Cohen & Lapata, 2018b) based on convolution neural network (CNN) is proposed with a constraint on topic distributions.

A hybrid extractive-abstractive summarization

In general, the human mind practices both extractive and abstractive approaches to summarize any document (Jing et al., 2000). Initially the significant parts in the document are identified and extracted. Further, based on the extracted content, decisions are made on what to ignore, paraphrase and transform them into precise and understandable summary. Such hybrid models can be split into three main categories (i) depending on attention for consistency; (ii) word-level evidence or statistical methods for generating summary; (iii) training the content generator using pseudo labels or other heuristics.

A hybrid summarization model (Hsu et al., 2018) that integrates the benefits of extractive and abstractive summarization is proposed. Sentence-level attention facilitates the regulation of word-level attention in a manner that tokens that are less attended in a sentence are less likely to be considered. In addition to this, an inconsistency loss function is used to indicate the inconsistency between two levels of attention.

Initial approaches involved PEGASUS model (Zhang et al., 2020) that was trained to summarize long-documents. As the model was much complicated and more topic specific, generalizability was compromised. Another challenge with this model is the inefficient context comprehension. To make the model better and cost-effective, most of the researchers have contributed to introduce new parameters which are derived weights from large-pre-trained models. To optimize the selection of the hyperparameters, Fast Fourier Transform (FFT) function is integrated to ignore redundancies in hidden sequence to accomplish discrete cosine transformation (DCT) (He et al., 2023).

This reduces the computational cost but failed to retain the benefits of various large pre-trained models like bidirectionality, transfer learning, etc., (Edunov, Baevski & Auli, 2019) The EASE model (Li et al., 2021) relies on the principle of information bottleneck (IB) that represents the understanding between the size of the extracted proof content and the knowledge provided for generating the final summary. Ravaut, Joty & Chen (2022) utilized a simple but effective extractive and abstractive hybrid model (Ravaut, Joty & Chen, 2022) where the first phase identifies the primary sentences and combines them to generate a summary.

Gap analysis

Most of the existing methods are task specific and involve numerous parameters for the conceptual summary generation but miss to consider the semantical relevance between the sentences of the input document. A document encompasses topic specific information which is more detailed and redundant. The redundant data must be removed and significant information has to be considered to generate an informative and concise summary. The key motive of our proposed model is to develop a contextually firm system that can learn the contexts from end-to-end. This is done by extracting contextual features from the sentences in the source document through which the informative sentences are identified. By taking those informative sentences as input, an abstractive summary is generated which is precise, informative and similar to the human generated summary. Although, the recent summarization techniques incorporate pre-trained models and large language models, it could not generate summary which is exactly similar to an expected summary. This affects the ROUGE score to a greater extent. For low resource languages, training cannot be performed effectively due to the unavailability of datasets with efficient reference summary. This research work relies on the semantic relevance of contextual features for the generation of summary and is compared with the recent summarization techniques that are focused on quality summary.

Methodology

Problem description

Summarization aims in generating a summary that is readable, relevant, simple and informative without any redundancy. Given an input document D, consisting of multiple sentences D={S1,S2,…,Sn}, where Si is the ith sentence in the document. Extractive summarization uses sentence selection Se1,Se2,…,Sen, with an assumption that the selected summary sentences comprise significant and relevant information from the document. This is accomplished by depicting the relationship between sentences. Z={Z1,Z2,…,Zm} represents ‘m’ the number of significant sentences extracted from source document. Extracted informative sentences (Z) are considered as input for generating an abstractive summary. Language generation model is incorporated to form new tokens and phrases to generate an abstractive summary (Y). Figure 1 illustrates the architecture of the proposed hybrid summarization model. The details of each phase are explained below.

Figure 1 Proposed hybrid summarization model.

The overall architecture of the proposed model is detailed. This figure explains the flow of the summarization model. Input text document is taken as input. The input is pre-processed to ignore special characters and irrelevant content before processing. Intermediate representations of each sentence are generated using embedding techniques. These representations are considered to extract informative sentences from the input document. These informative sentences are fed as input to language generator to generate abstractive summary.

Document pre-processing

In general, a document contains pictures, tables, references, symbolic representations, titles which will not contribute much to the semantic representation. Input document is broken down as a list of sentences which is termed as ‘Sentence Tokenization’. Few sentences in the document are semantically redundant and irrelevant, such elements can be ignored from the input document before being processed further. The position and size of the sentences plays a vital role in identifying the significance. Sentences which lie at the beginning of the document generally emphasize the knowledge about the document’s domain. Longer sentences hold more information whereas the shorter sentences have less information which means that the sentences that are long are given more priority. Pre-processing methods such as tokenization, ignoring short sentences, removal of symbols and other techniques are applied to the input document for analyzing and cleaning for further processing.

Sentence representation

Bidirectional Encoder Representation using Transformers (BERT) (Devlin et al., 2018) is a deep bidirectional model designed to generate representations from the unlabelled text. During the pre-processing phase, each input token is represented to the model using a specific set of rules. Each input embedding is a conjunction of position, segment and token embeddings. This deep bidirectional model acquires information from both directions during the phase of training. This model is pre-trained on two NLP tasks, masked language modelling (MLM) and next sentence prediction (NSP). This model is trained to predict masked/hidden words in the sentence. It is also trained to predict the following sentence of the given input sentence which deserves the knowledge of association between sentences. Being trained for these two tasks, this model holds the capacity to generate a conceptualized representation for each given input unit.

Sentence embedding is performed using Sentence Bidirectional Encoder Representation using Transformers (SBERT) (Radford et al., 2018) which is an alteration of BERT network that incorporates Siamese and triplet network to obtain contextually sensible vector representation of sentences. BERT involves a cross-encoder architecture and pair of sentences are fed as input to the transformer network to predict the target value which may not be suitable for several regression tasks because of possible associations. A general way to perform clustering is by associating each sentence to a vector space in such a way that semantically similar sentences lie nearer to each other. This led to the idea of picking an individual sentence as an input to BERT to obtain its embeddings. To improve the effectiveness of the embeddings in such an approach, SBERT is utilized. This network architecture empowers the computation of static length vectors for input sentences.

SBERT is fine-tuned on natural language inference (NLI) data to generate sentence embeddings that surpass other sentence embedding techniques. SBERT utilizes the pre-trained BERT and fine-tuning is done to produce efficient embeddings and minimize the training time. A pooling layer is appended on top of the BERT output to return static-sized embedding. A Siamese and triplet network (Schroff, Kalenichenko & Philbin, 2015) is built to upgrade the weights to create semantically meaningful embeddings. These embeddings facilitate semantic similarity computation using distance calculation metrics. The structure of the network relies on the type of training data (labelled or unlabeled) and is tested with the following objective functions.

Objective function for classification: The embeddings for an input sentence pair (u,v)is appended with the element-wise difference |u,v| and is multiplied with trainable weights Wt∈R3n×k,

(3) o=softmax(Wt(u,v,|u−v|))

Here n is the sentence embedding dimension and k is the label count. The cross-entropy loss is optimized.

Objective function for regression: The distance between two sentence embeddings is computed using a cosine similarity measure and Mean-squared-error loss is applied as the objective function.

Objective function for triplet network: Three sentences is considered for tuning the loss in the network. Initially, an anchor sentence (a) is picked. A positive sentence (p) having the same label as that of the anchor sentence is chosen and another sentence (n) having the other label is chosen. The triplet loss is estimated in a manner that the gap between ‘a’ and ‘p’ is less than the gap between ‘a’ and ‘n’. This estimation is utilized to tune the weights associated with the network. Mathematically,

(4) max(||Sa−Sp||−||Sa−Sn||+ϵ,0).

Here, Sa,SpandSn represents the sentence embeddings of sentences a, p and n respectively. ||.|| represents the distance measure and ϵ is the margin that projects that SpisatleastϵclosertoSpthanSn. In general Euclidean distance is applied to compute the distance with the margin ϵ set as 1 (ϵ = 1).

SBERT for regression generates the representations and performs distance estimation using the cosine similarity calculation method. The semantical representations generated by the BERT model analyzes the input on both token level and sentence level and so is capable of handling diversity on all aspects. In the proposed model the conceptual representation for the pair of pre-processed sentences is fetched before distance estimation and is considered as input for further processing. Steps incorporated in the generation of intermediate representation for each sentence are illustrated in Fig. 2.

Figure 2 Steps in generation of intermediate sentence representation.

This figure describes each step and various factors considered in the generation of intermediate representation for sentences.

Proposed hybrid summarization model

In general, a document comprises a greater number of sentences which makes the summary generation complex. To deal with this challenge, the proposed model integrates extractive and abstractive summarization techniques (Algorithm 1). To facilitate effective summarization, redundant and irrelevant information has to be ignored for the selection of significant sentences. A novel extractive summarization model is proposed to identify significant sentences from the source document. These extracted significant sentences are taken as input for generating an abstractive summary, that is similar to the manually created summary.

Algorithm 1 Hybrid summarization algorithm.

Input: Text Document (TD)	
Output: Extractive Summary (ES)	
Begin	
        /*Text pre-processing (TD)*/	
Expand contractions, sentence tokenization, ignore supplementary details, sentence position and sentence size tuning	
       return {S1, S2, …….., Sn};	
       /* Where, Sn = pre-processed_Sentences, n = number of pre-processed_Sentences in the document*/	
       /* Feature Extraction using SBERT {S1, S2, …….., Sn} */	
           SBERT:	
               {	
                input: pair of sentences	
                BERT: 24 layers of transformers,	
                mean pooling,	
                maps each sentence to semantically significant representation	
               };	
       return (N×E), {E1, E2, …….., En};	
       /* Where, E = embedding dimension, representation of n sentences = En = Data-points*/	
       /*Sentence Similarity using Manhattan distance metric {E1, E2, …….., En}*/	
       /*Where, distance between pair of sentences = distance between real valued features of sentence pair*/	
                Input = sentence pair (Sa,Sb)	
                Where, Sa = {ea1, ea2, …, ean} and Sb = {ab1, ab2,…, abn}	
                                         for (a←0 to n)	
                                              for (b←0 to n)	
                dist = |Sa − Sb| = |ea1 − eb1| + |ea2 − eb2| + |ea3 − eb3| + … |ean − ebn|	
                                   end for	
                              end for	
       return dist[a][b];	
       /* Matrix Generation ({E1, E2, …….., En}, dist)*/	
       /* Where, M = Matrix with n×n size;*/	
                           for i←1 to n do	
                               for j←1 to n do	
                                    insert dist[i][j] in M[i][j]	
                               end for	
                           end for	
          return M[i][j];	
       /*Identifying Contextually Similar Sentences from Matrix*/	
       Pick relevant sentence (exp_sum_size)	
       /*Where, exp_sum_size = user_expected_size of the summary to be generated*/	
       Select primary sentence from the input document	
         Generate rank matrix and include its respective sentences in the summary	
                 if (size (summary) < size(expected_summary))	
                 /*Where summary = collection of sentences from Matrix*/	
                       Select summary sentences from sentences in the matrix.	
                               if (corresponding sentences of matrix are not present in the summary)	
                                    Append the sentences in the summary	
                               end if	
                   end if	
                 return extractive Summary	
       /* Language Generation	
       Understand the context in the identified informative sentences.	
       The symmetric and invariant objective function is employed to rephrase the informative sentences.	
       Generate language to include all the content available in the extractive summary.	
       Summary similar to the manual summary will be generated.	
End	

Significant sentence extraction

A novel extractive summarization system takes the representation of each sentence in the input document as input and identifies the most informative sentences. The semantic relevancy between sentences is identified by estimating the distance between each sentence representations. Several distance calculation metrics (Huang, 2008) are available to estimate the relevancy between sentences. Manhattan distance estimation is proven to be effective in determining the distance between high-dimensional data (Aggarwal, Hinneburg & Keim, 2001). This works by calculating the covered distance while tracking a grid-like path between two points. Considering two high-dimensional points x=x1,x2…xi and y=y1,y2…yi,the Manhattan distance measurement is performed using,

(5) d=∑i=1n⁡|xi−yi|

where, n is the number of dimensions, xi and yi are high-dimensional data points (sentence representations) and d is the distance between two sentence representations. A greater distance value between two sentences indicates that the sentences are dissimilar. An adjacency matrix is generated with rows representing the sentence ids and columns representing the elements holding the distance value between two corresponding sentences. This matrix has 0’s in its diagonal value as the distance between a sentence with itself is zero. From this adjacency matrix, a rank matrix is derived that ranks each sentence based on its distance from other sentences. In a rank matrix, the rows represent the sentence ids and the columns represent the rank. While constructing the rank matrix, the diagonal 0’s is eliminated. During the generation of the rank matrix, the elements of each row in the distance matrix are sorted in ascending order and the values are replaced with their respective sentence ids. Each column in the rank matrix represents the level of closeness among sentences. Concerning the first column, the most frequent sentence id is considered the primary sentence which is highly relevant to all the other sentences. The row of the selected primary sentence id are taken as the summary sentences. The selection of summary sentences is based on the desired summary size. Finally, the summary sentences are arranged sequentially according to the given input document.

Consider an input passage having six sentences S={S1,S2,S3,S4,S5,S6}, whose representations are generated using Sentence BERT. The distance matrix is generated by computing the distance between sentences. The distance matrix having zeros in its diagonal shows that the distance between a sentence with itself is zero. From this distance matrix, a rank matrix is generated which ranks the sentences based on its similarity. In the rank matrix, the row represents the sentence ids and the elements in the row are arranged based on the closeness with the other sentences. In the above example from Fig. 3, for sentence ID 1, the elements are arranged as S2, S4, S3, S6 and S5. The S1 is not considered as the closeness to itself is 0. This reduces one column from the distance matrix while constructing the rank matrix. The size of the rank matrix is (numberofsentencesxnumberofrank) and the number of ranks estimated is (Numberofinputsentenes−1). The first column holds the sentence that is more close to each other sentences. The sentence ids that are most frequent in the first column are taken as the primary sentence, which is closer to all the other sentences. In situations where two different sentence ids occur the same number of times, track the second column and identify the sentence id that is most frequent. Each column in the rank matrix shows the level of closeness between sentences. Elements in selected row are sorted in ascending order and the distance values are replaced with their corresponding sentence ids. In the given example, the first column in the rank matrix S1 is most frequent, which means that, the sentence S1 is semantically similar to all the other sentences. Thus, S1 is considered as the primary sentence. The row of the selected sentence id indicates the summary sentences. Here, the elements in the row S1 is {S2,S4,S3,S6,S5}, which states the hierarchy in which these sentences are closer to S1. Selection of summary sentences from this set {S1,S2,S4,S3,S6,S5}, is based on the desired summary size. Finally, the summary sentences are arranged sequentially according to the given input document. These extracted summary sentences are considered as input for generating an abstractive summary. The sentences extracted from the input document based on the desired summary size are presented in Table 1.

Figure 3 A sample for significant sentence extraction.

A sample passage having six sentences is considered as input. The method applied on this sample input is to identify the informative sentences.

Table 1 Significant sentence selection outputs for the above-given sample.

Input document sentences	{S1,S2,S3,S4,S5,S6}	
Primary sentence ID	{S1}	
Sentences closer to primary sentence	{S2,S4,S3,S6,S5}	
Summary sentences along with primary sentence.	{S1,S2,S4,S3,S6,S5}	
25% of desired summary	{S1,S2,}	
50% of desired summary	{S1,S2,S4}	
80% of desired summary	{S1,S2,S3,S4,S6}	

Language generation

Followed by the extraction of a certain number of significant and semantically relevant sentences from the input document, a model that learns to originate an abstractive summary is incorporated. Here the significant sentences extracted are denoted as X ={X1,X2,X3,X4,X5,X6}, where Xiϵn represents the representations of the extracted sentences for which the abstractive summary has to be created. Two main challenges to be addressed during abstractive summarization are: (1) complete context has to be considered by the decoder during the prediction of words. (2) Exploiting pre-trained contextualized languages on the decoder phase to facilitate learning of summary representations, contextual interactions and modelling of languages collaboratively. To generate an abstractive summary, BERT based language generation model is incorporated (Zhang, Xu & Wang, 2019). This model utilizes a pre-trained language model during its encoder and decoder process and the training is performed end-to-end without manually extracted features. This model generates each word in the summary by exploring the contextual information from both sides. The functioning of language generation using BERT based model is illustrated in Fig. 4. Summary draft generation is rooted in the sequence-to-sequence network. Extracted significant sentences are considered as the input document for generating an abstractive summary. The model used for generating language has two phases, encoder and decoder. The encoder phase encodes the input document X ={X1,X2,…,Xm}, into intermediate representation R ={R1,R2,R3,…,Rm}. BERT model is used as the encoder which initially generates word embeddings for the input sequence and then computes document embeddings, which is the output of the encoder.

Figure 4 Steps in language generation.

The process of generating abstractive summary from the informative sentences is detailed. This figure describes the functional flow in the abstractive summary generation.

(6) R=BERT(X).

This representation R is given as input to the decoder for the generation of summary draft D ={D1,D2,…,D|d|}.The summary draft decoder comprises of word embeddings matrix generated by BERT to associate the output of the precedent summary draft into generated embedding vectors at every time step. As the entire input sequence is not fed as input to the decoder, the BERT model is not employed for the prediction of context vectors. An N-layer Transformer decoder is implemented to obtain the conditional probability P(D|R). The multi-head attention mechanism in the Transformer’s encoder-decoder facilitates the decoder to acquire smooth alignment between input document and summary. At each timestep, the prediction of output probability conditioned on the encoder’s representation and precedent output is estimated by the summary draft decoder. Each sequence generated is truncated at the occurrence of the ‘[PAD]’ token. The summary draft decoder integrates the output from the multihead attention decoder and the actual summary draft output. The learning objective of the decoder is to alleviate the negative likeliness of conditional probability. step

The bi-linear dot product of the final layer decoder output of the Transformer and the encoder output is initially estimated to compute the attention probability distribution over input document X. The choice of words in the summary is performed using a copy mechanism (Gu et al., 2016) based on the Transformer decoder to avoid including out-of-vocabulary tokens. Calculation of copying gate

(7) gtϵ[0,1]

that brings a smooth preference between choosing a word from an input document and building from the vocabulary. The weighted sum of copy probability and generation probability is computed to obtain the final probability of extended vocabulary V+X, where X is the out-of-vocabulary words from input document.

(8) gt=sigmoid(wg[et,s]+b)

where, Wg is a linear transformation with g indicating the token significance, et is the output from the decoder’s last layer, s is the output from the encoder, b is the bias and e^t manages the significant feature information is calculated as

(9) e^t=gtet

The refining process is to improve the efficiency of the decoder by applying BERT’s contextualized representations to ignore modifying the encoder for re-utilization. A word-level refined decoder is included in the decoder phase that takes generated summary draft as input and the refined summary is generated as output. This initially masks every word in the draft one after the other and is given as input to the BERT model for the generation of context vectors. A refined summary word is predicted by applying an N-layer Transformer decoder, similar to the draft decoder. At each time step, a specific word in the summary draft is masked, from which the refined word is predicted by taking other words as the input. Thus, the refined decoding stage generates a complete and consistent sequence with their pre-training processes. Thus, the language generation process works as follows: Initially, a summary draft is generated by the draft decoder based on the input document (extracted significant sentences). This summary draft is modified using the refine decoder. Each word in the summary draft is focused on refinement, as the execution is similar to the BERT pre-training process. With the capability of the contextual language model, the decoder can generate more fluent and complete sequences.

Experiments and results

Experimental data

Implementation of the proposed hybrid summarization model is performed using packages such as spacy, pandas, NumPy, sklearn and Transformers. The proposed model is applied to a benchmark dataset and its performance is assessed with ROUGE metrics implemented using PyRouge package. The details of the dataset and the performance metrics used to evaluate the generated summary are discussed below.

Dataset

The performance of the proposed model is examined on English language-based news dataset CNN/DailyMail. This dataset holds a huge volume of news articles developed by CNN and DailyMail news reporters. A non-anonymized version of this dataset (See, Liu & Manning, 2017) is purely designed for machine-reading understanding and question answering. The latter version of this dataset (Hermann et al., 2015) is compatible with extractive and abstractive summarization tasks. This constitutes textual news passages and the respective summary with the highlights. This includes 280,000 training samples, 13,368 verification samples and 11,490 test set samples approximately. Each news passage contains 40 sentences and 781 tokens on average. The news passages from this dataset are taken as input to the proposed hybrid summarization model and the summary is generated. The generated summary is then compared with the reference summary to assess the summary quality, which contributes to the evaluation of the hybrid summarization model.

Evaluation metrics

The summary quality is evaluated using ROUGE metrics (Lin, 2004), an extensively used quantified metric. ROUGE score is determined through the overlaying words among the generated and reference summaries.

(10) ROUGEn=∑sϵ[Referencesummaries]∑gramnϵSCountmatch(gramn)∑sϵ[Referencesummaries]∑gramnϵSCount(gramn).

Several variants of ROUGE such as ROUGE-1, ROUGE-2 and ROUGE-L are generally utilized for evaluation. Summary information is assessed using ROUGE-1 and ROUGE-2 and summary fluency is evaluated using ROUGE-L (Longest common subsequence). Assume the reference summary as R, where R={r1,r2,r3,…,ri} and ri is the ith word in the summary. System generated summary is represented as S, where S={s1,s2,s3,…,sj} and sj is the jth word in the summary. The estimation of precision, recall and F1-score of ROUGE-N is determined as follows.

(11) Precision=n(R∩S)n(S)

(12) Recall=n(R∩S)n(R)

(13) F1−Score=2×Precision×RecallPrecision+Recall.

Here, n represents the n-gram unit and n(X) shows the size of set X. If n = 1, the overlaying of unigram words is estimated. Overlaying of the longest matching sequence of words is estimated using ROUGE-L. F1-score of various ROUGE metrics is utilized for evaluating the system-generated summary.

Experimental results

Evaluation of extractive summarization model on CNN/DailyMail dataset

The novel significant sentence extraction phase in the proposed hybrid summarization model takes sentence representation as input and generates a distance matrix, from which the rank matrix is generated. The primary sentence (closer to all the remaining sentences) is selected from which the sentences to be included in the summary are selected. After arranging the extracted sentences in the input document sequence, a summary is generated based on the desired summary size. The quality assessment of different sized summary performed using ROUGE metrics and is shown in Table 2. When the summary size increases, the possibility of including most of the input sentences increases. Extracting all the informative sentences from the input document to generate the summary would generally make the summary size be half the size of the source document. The novel significant sentence selection method is compared with the existing standard extractive summarization model. The existing approaches that are known for their remarkable performance are detailed below.

Table 2 ROUGE score of various sized summary generated by significant sentence selection phase in proposed system.

Required summary size	ROUGE-1	ROUGE-2	ROUGE-L	
30%	38.52	17.83	37.32	
50%	42.81	19.17	38.24	
60%	46.95	21.32	40.51	
75%	49.21	22.54	42.26	

Leading Sentences (Lead-3) (Woodsend & Lapata, 2010) generates summary by electing the three significant sentences. Phrase-based ILP model (He et al., 2018) functions based on linear programming formulation that effectively merges phrases by examining significant features. Despite depending on shallow features, these two techniques provide significant results. Across several neural network-based models, NN-SE (Cheng & Lapata, 2016) employs a hierarchical document encoder and an extractor that functions based on the attention mechanism. SummaRuNNer (Nallapati, Zhai & Zhou, 2017) is a summarization model that functions based on a recurrent neural network. A neural network model that employs a hierarchical structured self-attention mechanism to generate sentence and document embeddings is HSSAS (Al-Sabahi, Zuping & Nadher, 2018). A neural network model that handles summarization tasks as a Contextual Bandit (CB) problem is BANDITSUM (Dong et al., 2018). This model takes a document as input and selects a sequence of informative sentences to be included in the summary. This function intends to increase the ROUGE score. Table 3 provides the ROUGE score of the system-generated summary of size half the input document that is compared with scores of exiting benchmark extractive summarization techniques executed on CNN/DailyMail dataset.

Table 3 Performance metrics for standard extractive summarization techniques.

MODEL	ROUGE-1	ROUGE-2	ROUGE-L	
LEAD-3 (Woodsend & Lapata, 2010)	39.2	15.7	35.5	
NN-SE (Cheng & Lapata, 2016)	35.4	13.3	35.6	
SummaRuNNer (Nallapati, Zhai & Zhou, 2017)	39.9	16.3	35.1	
HSSAS (Al-Sabahi, Zuping & Nadher, 2018)	42.3	17.8	37.6	
BANDITSUM (Dong et al., 2018)	41.5	18.7	37.6	
Proposed significant sentence selection	42.81	19.17	38.24	

Evaluation of abstractive summarization model on CNN/DailyMail dataset

The language generation phase (Zhang, Xu & Wang, 2019) in the proposed model is applied to CNN/DailyMail dataset and the performance of various recent abstractive summarization techniques are compared with it and the results are tabulated in Table 4.

Table 4 Performance metrics for standard abstractive summarization techniques.

MODEL	ROUGE-1	ROUGE-2	ROUGE-L	
Pointer Generator + Coverage (See, Liu & Manning, 2017)	39.53	17.28	36.38	
ML+ RL + intra-attention (Paulus, Xiong & Socher, 2017)	39.87	15.82	36.90	
Inconsistency Loss (Hsu et al., 2018)	40.68	17.97	37.13	
Bottom-Up Summarization (Gehrmann, Deng & Rush, 2018)	41.22	18.68	38.34	
Seq2Seq + atten (Bahdanau, Cho & Bengio, 2014)	31.34	11.79	38.10	
words-lv2k-temp-att (Nallapati et al., 2016)	35.46	13.30	32.65	
SummaRuNNer-abs (Nallapati, Zhai & Zhou, 2017)	37.50	14.50	33.40	
Pointer-generator (See, Liu & Manning, 2017)	36.44	15.66	33.42	
RL+ML (Paulus, Xiong & Socher, 2017)	39.87	15.82	36.90	

The approaches with which the proposed model is compared are detailed below: Seq2Seq + atten (Bahdanau, Cho & Bengio, 2014):- Abstractive summarization is performed using sequence-to-sequence technique incorporated with attention mechanism.

words-lv2k-temp-att (Nallapati et al., 2016):- Encoder-Decoder based model that utilizes temporal attention mechanism to track the previous decoder attention weight and retains the redundant parts in the preceding sequence.

SummaRuNNer-abs (Nallapati, Zhai & Zhou, 2017):- Extractive summarization is done using Recurrent Neural Network-based model and is further fine-tuned to perform abstractive summarization.

pointer-generator (See, Liu & Manning, 2017):- Sequence-to-Sequence model based on an attention mechanism that is rooted in a hybrid-pointer generator to solve rare words and Out-Of-Vocabulary problems.

pointer-generator + coverage (See, Liu & Manning, 2017):- improvised pointer-generator model by including a coverage mechanism to ignore redundancy.

RL + ML (Paulus, Xiong & Socher, 2017):- A Neural Network model that has an intra-attention mechanism and a novel training method for generating an abstractive summary.

Tables 3 and 4 prove that the performance of two phases (significant sentence selection and language generation) in the proposed model being evaluated individually outperforms several extractive and abstractive summarization techniques.

Evaluation of proposed hybrid summarization model on DUC2004 dataset

The proposed model is trained on CNN/DailyMail dataset and is tested on DUC2004 (Over & Yen, 2004). DUC2004 dataset contains 500 documents and four references summary generated by experts. ROUGE metrics is used to evaluate the proposed model. The proposed model is compared with some fundamental and state-of-the-art models and is given in Table 5.

Table 5 Comparative results of proposed model with recent abstractive summarization model on DUC2004 dataset.

Models	ROUGE-1	ROUGE-2	ROUGE-L	
TOPIARY (Zajic, Dorr & Schwartz, 2004)	25.12	6.46	20.12	
MOSES+ (Rush, Chopra & Weston, 2015)	26.50	8.13	22.85	
RAS-LSTM (Chopra et al., 2016)	27.41	7.69	23.06	
RAS-Elman (Chopra et al., 2016)	28.97	8.26	24.06	
LenEmb (Kikuchi et al., 2016)	26.73	8.39	23.88	
lvt2k-lsen (Nallapati, Zhai & Zhou, 2017)	28.35	9.46	24.59	
lvt25-lsen (Nallapati, Zhai & Zhou, 2017)	28.61	9.42	25.24	
Proposed model	31.82	10.81	27.51	

TOPIARY (Zajic, Dorr & Schwartz, 2004) is the efficient model on DUC2004 for summarizing text document. This method is the integration of linguistic based transformation and unsupervised model for topic detection to compress the text to generate summary.

MOSES+ (Rush, Chopra & Weston, 2015) uses statistical approach for machine translation in phrase level that are trained on for generating summaries. Deletion rules are applied on the phrase table to enhance the performance. BERT technique is incorporated to upgrade the summary quality.

RAS-LSTM and RAS-Elman (Chopra et al., 2016) methods analyze each word and its position and then applies convolutional encoders to handle the information. Decoding is performed using RNN in RAS-Elman and RAS-LSTM uses LSTM for decoding.

LenEmb (Kikuchi et al., 2016) model determines the expected summary size by assessing the size of the embedding vector as the input.

Lvt2k-lsent and lvt5k-lsent (Nallapati, Zhai & Zhou, 2017) introduce a strategy to define the size of the vocabulary to enhance the training efficiency.

The results on the DUC2004 dataset are tabulated in Table. The proposed model yields better outcomes for all the ROUGE metrics. lvt2l-lsen and lvt5l-lsen (Chopra et al., 2016) involve features such as parts-of-speech tags, named entity tags and TF-IDF for the representations. Extraction of such features is time-consuming and complex. In the proposed work, the BERT model is used to generate intermediate representations. This model is pre-trained for Masked token identification and Next Sentence Prediction. Thus, this model can consider features on every level such as lexical, syntactic, semantic, discourse and pragmatic. Such features used for representing each sentence facilitate effective estimation of semantic relevance. This makes the extractive summarization efficient thereby leading to an informative and quantitative abstractive summary.

The reference summary in DUC2004 dataset consists of only one sentence for each document. Whereas in CNN/DailyMail dataset the highlights are available as summary which contains all the information in the input article. Thus, the ROUGE Score obtained for CNN/DailyMail dataset is higher compared to the DUC2004 dataset. However, the ROUGE score of the proposed model on both the dataset outperforms the existing systems.

The effectiveness of the proposed summarization model can be further increased by analyzing each sentence and processing them. Rather than feeding the extractive summary as such to the language generation phase, the sentences in the extractive summary can be joined together. This will further help in the generation of coherent and informative abstractive summary.

Conclusion and future works

In this article, an effective hybrid summarization model that employs both abstractive and extractive summarization is presented. A novel extractive summarization model is proposed in which a primary sentence that is closer to all the other sentences is selected. Summary sentences are chosen based on the semantic similarity with the primary sentence. The extracted summary sentences are given as summary to the abstractive summarization phase to generate a summary that is fluent, readable and consistent. The extensive implementation of the proposed model on CNN/DailyMail dataset achieves improved results when compared to recent standard summarization models.

Amongst several native languages, Tamil is morphologically rich language and there is a lot of scope for research. In general, the generation of semantic representation supports more in the effective identification of informative sentences. Such intermediate representation for sentences in Tamil is challenging as the pre-trained model that can extract context from the input is still not up to the mark. Thus, this research work can be extended to focus on identifying informative sentences from the Tamil language document by extracting the semantics from the document. Due to the lack of dataset with summary ground truth, evaluating the quality of the summary is still challenging. Thus, this research work can also be extended in developing a dataset for training a model for generating Tamil summary and metrics to evaluate the generated summary.

Supplemental Information

Supplemental Information 1 Process of extracting significant sentences.

This explains the steps involved in the extraction of informative sentences for the generation of extractive summary.

Supplemental Information 2 Deep Intermediate Representation of Sentences.

The functional flow which represents the generation of intermediate representation for each sentence using BERT is detailed. [CLS] label is appended in the beginning of each sentence and two sentences are separated using [SEP] token. Each token in the sentence is considered as input to the embedding layer. These n dimensional vectors are given as input for the pooling layer to generate representation for each sentence.

Supplemental Information 3 Raw data.

Supplemental Information 4 Order of execution.

Additional Information and Declarations

Competing Interests

Author Contributions

Data Availability

The authors declare that they have no competing interests.

Divya S. conceived and designed the experiments, performed the experiments, performed the computation work, prepared figures and/or tables, and approved the final draft.

Sripriya N. conceived and designed the experiments, performed the experiments, performed the computation work, prepared figures and/or tables, and approved the final draft.

J. Andrew analyzed the data, authored or reviewed drafts of the article, and approved the final draft.

Manuel Mazzara analyzed the data, authored or reviewed drafts of the article, and approved the final draft.

The following information was supplied regarding data availability:

The raw data is available in the Supplemental Files.

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
