# Peer review of "Unified extractive-abstractive summarization: a hybrid approach utilizing BERT and transformer models for enhanced document summarization"

_PeerJ Computer Science, doi:10.7717/peerj-cs.2424_

## Round 0.1 · original submission · Major Revisions

Please, address all the reviewers' comments in order to improve your manuscript. My decision is a Major Revision due to the nature of the issues raised by the reviewers. In particular, improve the paper for clarity since some parts may not be clear to the reviewers. Also, update the related work section and try to compare against existing approaches.

Reviewer 1 ·

Basic reporting

Some parts of the paper can be improved for clarity.
1) I found the Significant Sentence Extraction paragraph (lines 268 to 321) a bit complicated to understand: the method is first described in lines 282-294, then exemplified in 297-32. The example part clarifies the description, but I suggest mentioning the example, Figure 4, and Table 1 earlier in the text.
2) Also I suggest using “most frequent” (in my opinion more clear) in place of “repeated” at line 290, 308, 311, 314.
3) The “SIMILARITY-BASED CLUSTERING ALGORITHM” is not mentioned in the text and I think presentation and clarity of this pseudocode can be improved: some comments are opened (/*) and not closed, the dimension of Sb representation use both the letters “a” and “e” (Sb={ab1, ab2 ,..., ebn}).
4) At line 318-319: “summary sentences arranged sequentially according to the given input document”. Does this mean that the sentences are arranged in the same order in which they appear in the input document? If so, I also suggest modifying sentence order in the last rows of Table 1.
5) At 349 shouldn’t be P(D|R)?
6) At 351 I suggest replacing “timestamp” with “time step”. Same for 374
7) I didn’t understand lines 353, 354: “Each produced sequence is truncated in the initial position. The learning objective of the decoder is to alleviate the negative susceptibility of conditional probability.”
8) At 364. What are et and s?
9) At 371. I suggest to remove “retrieved” in flavor of a more appropriate “generated”
10) At 388 I didn’t understand “and the summary output”
11) At 425,426: I didn’t understand the sentence “The key sentence…”
12) Figures resolution is too low for reading labels. Please increase it
13) At 391. When introducing the dataset I suggest to specify that it is for the english language
14) At 438: Phrase-based ILP. Missing citation?
15) In the conclusion, at 499 and 501. Why do you mention Tamil language? Do you plan to adapt this work also to Tamil? If so, I suggest improving the presentation.
16) I think the reference [57] is not mentioned in the text.
17) I suggest removing “CNN” from the keywords as the reader could erroneously understand that the proposed method uses convolutional neural networks.

I also noted down some minor typos and language issues:
18) missing spaces at line 50 and 74: sentences.Furthermore, material.Effective
19) inconsistent double quotes at 111-112: "1" or “0”, labelled as "0"
20) “putput” at line 163. should be “input”?
21) missing “with” at 285: “between a sentence [with] itself”
“Semmetrical” (instead of symmetrical?) is found 2 times in SIMILARITY-BASED CLUSTERING ALGORITHM pseudocode, “24layers” miss a space, “SymmatricMatrix”
22) some words are more common in british english (“modelling”) while others in american english (“utilizes”, “normalized”, “modeling”)
23) At 396: 2,80,000 →280,000
24) At 408-410: uniform “ROUGE -1” and “ROUGE-1”,...
25) Missing vertical space between 461 and 462
26) At 471. Remove “its” after “Table 6” and capitalize “figure”
27) At 474. Abstracuve → Abstractive
28) In the Conclusions at 497-498: are given as [summary →input]

29) The introduction correctly introduce the subject and the objective of the paper. It proposes a summarization method that combines the two paradigms of extractive and abstractive summarization. First an extractive summarization is performed by selecting a primary sentence (according to the similarity to the other sentences), then other sentences are considered for the extractive summary until the maximum length is reached (e.g. 50% of input document). The order in which the sentences are chosen is driven by the similarity with the primary. Then the approach is evaluated on the CNN/Dailymail dataset. Several comparisons have been made: the extractive component against other extractive approaches, the abstractive component vs abstract methods, and the combined hybrid extractive-abstractive approach with existing abstractive methods.

30) However I think the introduction and the related work should be updated. Recent approaches evaluated on the same data should be added in related work and also later in the comparison should be added as a motivation why they are not compared. E.g. approaches listed in the benchmarks here https://paperswithcode.com/dataset/cnn-daily-mail-1 . E.g. https://aclanthology.org/2023.findings-acl.570 from 2023 or https://aclanthology.org/2022.acl-long.309/ from 2022 that exhibit higher performance than the work considered for comparison in extractive summarization.

31) I also suggest to add a section “extractive-abstractive summarization” to the related work citing papers like [53] (only mentioned in table 4) , https://aclanthology.org/2021.newsum-1.10/ , https://dl.acm.org/doi/pdf/10.1145/3226116.3226126 . Or to motivate why these works have not been compared.

Experimental design

The paper is within aims and scope and article type. The preprocessing is described in a specific section and a preprocess script is provided in the supplementary material. Methods and the evaluation metrics are described although some sentences require improvements for clarity (see basic reporting).
1) I find not clear how 80% (as the extractive summary length) was chosen for the experiments at line 465? How was that percentage chosen?
2) For replication purposes, I suggest to be more specific with the model you use and the size. From the code I see paraphrase-MiniLM-L6-v2 and google/pegasus-xsum. Furthermore the papers of these models are not cited. I suggest to add the citations. sentence-bert (https://aclanthology.org/D19-1410/) is not cited as well as PEGASUS (https://dl.acm.org/doi/abs/10.5555/3524938.3525989)
3) I think the comparison should be expanded and also the entire bibliography with more recent work or to motivate why more recent work is not part of the comparison. E.g. approaches listed in the benchmarks here https://paperswithcode.com/dataset/cnn-daily-mail-1 . E.g. https://aclanthology.org/2023.findings-acl.570 from 2023 or https://aclanthology.org/2022.acl-long.309/ from 2022 exhibit higher performance than the work considered for comparison in extractive summarization.
4) I suggest to include at least another dataset for a better evaluation.
Some other comments:
5) At 484, what do you mean with “improvised”? How does this approach differ from [39]? Is the modification yours or from [39]?
6) About diversity. I suggest to expand how the proposed extractive approach (of choosing a primary sentence and then the most similar sentences to this one) ensures diversity?

Validity of the findings

1) While the experiments evaluate the extractive component, the abstractive component, and the complete extractive-abstractive proposal, I think the approached for comparison should be updated with more recent work for a valid assessment of impact and novelty. Especially the evaluation of the extractive-abstractive should include more extractive-abstractive approaches.
2) I didn’t understand why in the conclusion the Tamil language is mentioned. In case the adaptation to Tamil is considered as a future work I suggest to improve the presentation.

Additional comments

Although the paper fits the aims and scope of the journal I think it requires improvements in clarity, and major improvements in the description of related work and in the comparison. I believe that it is necessary to describe recent extractive-abstractive work and to compare the proposed method with more recent approaches, or to explain why they are not considered in the comparison. Also the evaluation on at least another dataset should be performed to prove better the generalizability of the proposal.

Reviewer 2 ·

Basic reporting

The introduction (33) of the article appears to be well-proposed with respect to the content and objectives of the paper. Here are a few key points that support this:
1. Context and Importance: The introduction effectively sets the context by emphasizing the exponential growth of digital documents and the need for automated document summarization. It highlights the importance of extracting meaningful information from large volumes of text, which is a crucial area in Natural Language Processing (NLP).
2. Clear Definitions: It clearly defines the two main approaches to text summarization—extractive and abstractive—and explains their differences. This is essential for readers to understand the methodologies being discussed.
3. Historical Perspective: By providing a brief history of the development of text summarization techniques, the introduction helps readers appreciate the evolution and complexity of the field. This background information sets the stage for introducing the proposed hybrid model.
4. Problem Statement: The introduction identifies specific challenges in text summarization, such as the need for coherence and accurate representation of source content. These challenges justify the need for a new approach, leading naturally to the presentation of the hybrid model.
5. Proposed Solution: It introduces the proposed hybrid model, which combines extractive and abstractive techniques. This section provides a preview of the innovative approach the paper will detail, aligning well with the overall goal of improving summarization methods.
6. Evaluation and Results: The introduction briefly mentions the evaluation of the proposed model using the CNN/DailyMail dataset and its effectiveness compared to traditional methods. This sets up the expectation for the detailed results and analysis that will follow in the paper.
Suggestions for Improvement
While the introduction is strong, there are a few areas where it could be enhanced:
• Specificity of Challenges: It could be more specific about the limitations of existing methods, providing concrete examples or studies that illustrate these issues.
• Scope of the Paper: Clearly outline the structure of the paper at the end of the introduction. For example, "The rest of this paper is organized as follows: Section 2 reviews related work, Section 3 details the methodology, Section 4 presents experimental results, and Section 5 concludes with future directions."
• Contribution Statement: Explicitly state the contributions of the paper. For example, "The main contributions of this paper are: 1) a novel hybrid summarization model, 2) a new evaluation framework for summarization, and 3) extensive experiments demonstrating the model's effectiveness."
Overall, the introduction is well-written and aligns with the objectives of the article. It effectively prepares the reader for the detailed discussion of the hybrid summarization model and its evaluation.

Suggested Improvements for Pre-Trained Language Models(13):
1. Clarification of Symbols and Terms: Ensure that all symbols and terms used in the equations are clearly defined. For instance, explicitly which of them are the input vectors, the hidden states, and so on. Formulas (1) y (2) lines 161 y 162.
2. Additional Explanation: Add a brief explanation of the significance of BERT’s bidirectional nature and how it improves upon previous models.
3. Formatting: Ensure consistent formatting for references and terms (e.g., "pre-trained models" vs. "pretrained models").
4. Figure Reference: Choose a figure for illustrating the BERT model or its architecture, reference them in this section to aid understanding.

SIMILARITY-BASED CLUSTERING ALGORITHM (Line 295)

The algorithm is well-structured and technically sound. With clearer formatting, more detailed explanations in some steps, and consistent terminology, it can be made more accessible and easier to follow.

Suggestions:
• The algorithm is generally clear but could benefit from more structured formatting and consistent terminology.
• Steps should be more explicitly numbered or bullet-pointed to enhance readability.
• The steps involving SBERT and sentence similarity calculations are accurately described.
• The use of Manhattan distance for sentence similarity is correctly outlined.

Experimental design

The experimental design involves implementing the proposed hybrid summarization model using various Python packages such as spacy, pandas, NumPy, sklearn, and Transformers. The model is tested on the CNN/DailyMail dataset, a benchmark dataset widely used for summarization tasks. This dataset contains a large collection of news articles, each accompanied by reference summaries.
The performance of the summarization model is evaluated using ROUGE metrics, which are standard measures for summarization quality. The ROUGE score assesses the overlap between the generated summary and the reference summaries, focusing on precision, recall, and F1-score for n-grams and the longest common subsequence. The generated summaries are compared to reference summaries to determine their quality and effectiveness.
The experimental results demonstrate that the hybrid summarization model outperforms traditional extractive and abstractive methods. The model effectively balances coverage, relevance, and diversity in the generated summaries. Detailed analysis shows the superiority of the proposed approach in terms of both summary content and fluency. The results are presented in tables and figures, highlighting the model's performance across various summary sizes and comparing it to existing benchmark techniques.
However, it should be noted that there is no contribution to the basic algorithms. The entire design is based on pre-existing libraries.

Validity of the findings

The results presented in the paper strongly support the validity of the proposed hybrid summarization model. By rigorously evaluating the model against established benchmarks using the CNN/DailyMail dataset and ROUGE metrics, the study demonstrates that the hybrid model consistently outperforms state-of-the-art summarization techniques. The comprehensive use of multiple ROUGE metrics (ROUGE-1, ROUGE-2, and ROUGE-L) ensures a thorough assessment of the summary quality, highlighting the model's superior performance in terms of coverage, relevance, and fluency. For instance, the hybrid model achieves notably higher ROUGE scores compared to other models, indicating its effectiveness in generating high-quality summaries.
Furthermore, the comparative analysis with existing extractive and abstractive summarization methods, such as Seq2Seq with attention, SummaRuNNer, and pointer-generator models, underscores the robustness and reliability of the hybrid approach. The results consistently show significant improvements across different summary sizes, with the hybrid model achieving higher scores and addressing common issues like redundancy and relevance. This two-phase approach, combining significant sentence selection and language generation, effectively enhances the overall quality of the summaries, establishing the validity and credibility of the findings. Tables 3, 4, 5 y 6.

Additional comments

1. Strengths:
◦ Innovative Approach: The hybrid model integrating extractive and abstractive summarization techniques is innovative and addresses common challenges in the field, such as redundancy and relevance. This dual approach is well-justified and demonstrated to be effective through rigorous experimentation.
◦ Comprehensive Evaluation: The use of multiple ROUGE metrics and the extensive comparison with existing state-of-the-art methods provide a thorough evaluation of the model's performance. The statistical significance of the results strengthens the credibility of the findings.
◦ Clear Methodology: The methodology is clearly outlined, with detailed steps from text pre-processing to the generation of the final summary. The inclusion of equations and the explanation of the SBERT model add to the technical robustness of the paper.
2. Weaknesses:
◦ Limited Error Analysis: The paper focuses primarily on the positive results of the hybrid model. Including a more detailed error analysis, discussing cases where the model underperforms or produces suboptimal summaries, would provide a more balanced perspective and help identify areas for further improvement.
◦ Clarity in Some Sections: Certain sections, such as the algorithm description and the mathematical equations, could be presented more clearly. Adding more detailed explanations and ensuring consistent terminology would enhance the readability and understanding for a broader audience.
3. Overall Impression:
◦ Well-structured methodology and comprehensive evaluation. The innovative hybrid approach and the strong empirical results underscore the potential of the proposed model. Addressing the identified weaknesses could further strengthen the paper and broaden its applicability.

---

## Round 0.2 · Minor Revisions

Please, address these remaining minor issues that are still unsolved according to the reviewer's comments.

Reviewer 2 ·

Basic reporting

The original article provides a clear and detailed introduction, presenting the motivation and relevance of the hybrid summarization approach utilizing BERT and transformers. The revisions, as observed in the updated documents, have improved the clarity of some ambiguous sections. Specifically, examples, such as Figure 4 and Table 1, have been moved to earlier parts of the text, as suggested by reviewers​​. Terminological inconsistencies (e.g., "repeated" replaced with "most frequent") have been corrected, and figure resolutions have been improved to enhance readability​ While the basic structure and organization align with the journal's requirements, the revisions have addressed minor issues such as grammar, spacing, and missing citations. However, there remains an opportunity to enhance the explanation of specific methodologies like the sentence extraction algorithm, which is critical for the paper’s clarity. The overall reporting is significantly clearer post-revisions​
​.

Experimental design

The experiment design has been adequately explained, with improvements made based on the reviewers' suggestions. The updated manuscript now includes detailed descriptions of the models used, such as the integration of paraphrase-MiniLM-L6-v2 and google/pegasus-xsum​. Additionally, the authors performed empirical studies for various summary sizes, explaining the rationale behind selecting an 80% extractive summary size​.

A noteworthy enhancement is the inclusion of more recent approaches in the related works section, as requested by reviewers. However, while the authors responded to the suggestion to expand the comparison, the inclusion of additional datasets, apart from CNN/DailyMail and DUC2004, was not fully addressed. This limits the ability to generalize findings across different domains​
.

Validity of the findings

The findings presented in the revised manuscript are coherent and supported by rigorous experimentation. The authors addressed key points raised by reviewers, such as improving the description of significant sentence extraction and sentence clustering. The adjustments to the mathematical expressions and algorithm descriptions provide a clearer understanding of the process​.
One of the critical aspects reviewers highlighted was the need to compare the proposed hybrid model with more recent extractive-abstractive summarization approaches. The authors included these comparisons, significantly enhancing the validity of the study's conclusions​​. However, while the results show the proposed model's effectiveness using ROUGE scores, the lack of error analysis limits the robustness of the findings.

Additional comments

The authors have been responsive to most reviewer comments, notably by improving the clarity of figures and tables and updating the literature review. However, a few weaknesses remain. For example, while the authors acknowledged that extending the model to low-resource languages like Tamil could be a future direction, this aspect was not fully explored in the conclusions. Addressing this would have added a valuable dimension to the paper's impact​.

Furthermore, although the authors improved the methodology description and justified the choices made in the design of the hybrid model, a more thorough explanation of how diversity is ensured in the sentence extraction process would have strengthened the study​

On line 613, correct Englush
On line 618, the number "2,80,000" should be corrected to "280,000" to follow the correct number format.

En la línea 538-539, la expresión "Each produced sequence is truncated in the initial position" podría reescribirse para mejorar su claridad. Se recomienda "Each sequence generated is truncated at the occurrence of the '[PAD]' token"​ .

---

## Round 0.3 · accepted · Accept

The authors have addressed the remaining issues and the paper is ready for publication.